# *GmNMHC5*, A Neoteric Positive Transcription Factor of Flowering and Maturity in Soybean

**DOI:** 10.3390/plants9060792

**Published:** 2020-06-25

**Authors:** Wenting Wang, Zhili Wang, Wensheng Hou, Li Chen, Bingjun Jiang, Wei Liu, Yongjun Feng, Cunxiang Wu

**Affiliations:** 1School of Life Science, Beijing Institute of Technology, Beijing 100081, China; wangwt01@163.com (W.W.); zhiliwang0804@sina.cn (Z.W.); 2MOA Key Laboratory of Soybean Biology (Beijing), Institute of Crop Science, The Chinese Academy of Agricultural Sciences, Beijing 100081, China; houwensheng@caas.cn (W.H.); chenli01@caas.cn (L.C.); jiangbingjun@caas.cn (B.J.); hnaulw@126.com (W.L.)

**Keywords:** *Glycine max*, *GmNMHC5*, flowering regulation, CRISPR-Cas9

## Abstract

The soybean (*Glycine max* (L.) Merr.) is an important oil and food crop. Its growth and development is regulated by complex genetic networks, and there are still many genes with unknown functions in regulation pathways. In this study, *GmNMHC5*, a member of the MADS-box protein family, was found to promote flowering and maturity in the soybean. Gene expression profiling in transgenic plants confirmed that the 35S:*GmNMHC5* T3 generation had early flowering and precocity. We used CRISPR-Cas9 to edit *GmNMHC5* and found that late flowering and maturity occurred in *Gmnmhc5* lines with stable inheritance. Remarkably, in the 35S:*GmNMHC5* plants, the expression of flowering inhibitors *GmFT1a* and *GmFT4* was inhibited. In addition, overexpression of *GmNMHC5* in *ft-10* (a late flowering *Arabidopsis thaliana* mutant lacking *Flowering Locus T* (*FT*) function) rescued the extremely late-flowering phenotype of the mutant *A. thaliana.* These results suggest that *GmNMHC5* is a positive transcription factor of flowering and maturity in the soybean, which has a close relationship with *FT* homologs in the flowering regulation pathway. This discovery provides new ideas for the improvement of the flowering regulation network, and can also provide guidance for future breeding work.

## 1. Introduction

The MADS-box family is an important transcriptional regulator in plants. Members of this family have a similar secondary structure including the N-terminal DNA-binding MADS-box domain. The MADS-box domain is followed by an intervening region, the K-box, involved in protein–protein interactions, and the C-terminus, in which the divergence among members is greater [1]. MADS-box family proteins play a wide range of functions in plants, especially in the regulation of flowering time and the development of various reproductive organs [2,3,4]. The *Arabidopsis* MADS-box genes, e.g., AGAMOUS-LIKE 20 (AGL20), SUPPRESSOR OF OVEREXPRESSION OF CONSTANS 1 (SOC1), and AGAMOUS-LIKE 28 (AGL28), positively regulate the flowering process [5,6,7]. In turn, AGAMOUS-LIKE 18 (AGL18) negatively regulates flowering [8].

Some MADS-box genes are also involved in the regulation of root development. *GmNMH7,* a MADS-box transcription factor (TF), was found to inhibit root development and nodulation of the soybean [9]. ARABIDOPSIS NITRATE REGULATED 1 (ANR1) functions nutrient response in the roots and controls lateral root elongation in response to nitrate [10]. Recent studies have shown that AGL17 is also involved in the regulation of flowering time, which can promote the flowering of *Arabidopsis* by promoting the expression of *LFY* and *AP1* [11]. Early research has shown that *nmhC5*, which is orthologous to root-expressed AGL17 subfamily proteins in *Arabidopsis*, forms homodimers and performs its functions by binding to a CArG consensus sequence in vitro [12]. *GmNMHC5* was originally cloned from soybean (*Glycine max*) root nodules and was confirmed to promote the growth of soybean lateral root and root nodules using the soybean root transformation system [13].

The entire growth cycle of plants is influenced by the external environment and regulated by internal factors, of which the transformation from vegetative growth to reproductive growth is particularly important for determining the flowering time of plants. In most cases, how these genes act in regulating flowering time is not clear, but fundamental insights into the mechanisms underlying the transformation have been presented in recent years [14]. The soybean is a typical short-day plant, and its flowering process is strictly regulated by photoperiod. Its flowering process can be reversed when switching from short-day (SD) to long-day (LD) conditions [15]. Recent studies have revealed the maturity gene *E9* is *FT2a*, an ortholog of *Arabidopsis FLOWERING LOCUS T* [16]. Located downstream of the complex flowering regulation network, the *Flowering Locus T* (*FT*) homologs play very important regulatory roles, and have become a hotspot in the field of plant research.

Many studies suggest that *FT* has a central position in mediating the floral transition, being induced and transported to the apical meristem from leaves to perform functions [17,18]. Ten *FT* homologous genes have been identified in the soybean, but their roles in flowering regulation are different. *GmFT2a* and *GmFT5a* in soybeans were shown to be positive promoting factors of the flowering process [19,20,21,22]. *GmFT4* was shown to delay flowering in *Arabidopsis* [23]. In a recent study, *GmFT1a* was found to inhibit the flowering of the soybean [24]. *FT* was also revealed to be an important integrating factor in the photoperiodic pathway that was regulated by *CONSTANS (CO)* [25] and the *FT*/*FD* protein complex [26].

In this study, we found that overexpression of *GmNMHC5* promoted early flowering and early maturation of the soybean, and the CRISPR-Cas9-edited *Gmnmhc5* lines showed an obvious late flowering phenotype, with the mature stage being delayed accordingly. We also created *Arabidopsis* transgenic lines 35S:*GmNMHC5-ft-10* and used them to reveal the relationship between *GmNMHC5* and *FT* in regulating the flowering pathway. Taken together, we propose that *GmNMHC5* is a positive regulator of flowering, and the soybean has developed a balanced pathway to control flowering through coordinated regulation between the flowering promoter *GmNMHC5* and repressors *GmFT1a* and *GmFT4*.

## 2. Results

### 2.1. Overexpression of GmNMHC5 Significantly Promoted Flowering in Soybeans

To investigate the function of *GmNMHC5*, a constructed plasmid containing the *GmNMHC5* CDS driven by the CaMV 35S promoter was transformed into the soybean Jack cultivar at mid-maturity. In the T2 generation, we compared the flowering time between *GmNMHC5* overexpression transgenic lines and WT plants under natural conditions (summer) in Beijing, China. We found that *GmNMHC5* overexpression in homozygous lines promoted flowering. The T3 progeny of the homozygous *GmNMHC5*-T3#5, *GmNMHC5*-T3#25, and *GmNMHC5*-T3#32 lines were grown under both LD and SD photoperiodic conditions, and the genotypes were subsequently examined to confirm that *GmNMHC5* was stably overexpressed when inherited from the T2 generation (data not shown).

In terms of flowering, 35S:*GmNMHC5* mutants showed early flowering under both LD and SD (Figure 1a,b). The WT plants flowered at 35.4 days after emergence (DAE) under SD conditions, while the *GmNMHC5* transgenic lines flowered at 22.6 DAE (line 5), 21.0 DAE (line 25), and 21.7 DAE (line 32). Under LD conditions, the WT plants flowered at 46 DAE, while the flowering dates of the three transgenic lines were 38.8 DAE (line 5), 39.2 DAE (line 25), and 37.9 DAE (line 32). Through statistical analysis of sufficient samples, we confirmed that *GmNMHC5* promoted soybean flowering under both LD and SD conditions (Figure 1d). At the same time, the *GmNMHC5* overexpression plants also showed precocious maturation (Figure 1c,e). Under SD conditions, *GmNMHC5* transgenic strains were matured at 44.0 DAE (line 5), 45.0 DAE (line 25), and 45.4 DAE (line 32), respectively, while the WT matured at 55.4 DAE. Under LD conditions, WT plants matured at 69.8 DAE, and the maturation time of the three transgenic strains was 61.1 DAE (line 5), 59.4 DAE (line 25), and 55.4 DAE (line 32).

### 2.2. Flower Development of the 35S:GmNMHC5 Lines Occurred Earlier than That of the Wild-Type Lines

Paraffin sections of the wild-type and 35S:*GmNMHC5* materials in different periods were made to compare their anatomical structure. Under the same growth conditions, the flower primordia structure of the 35S:*GmNMHC5* lines appeared at 30 DAE, while no flower structure was observed in the WT. At 36 DAE, the wild-type flower primordia had formed. At 46 DAE, the flower structures of wild-type plants was complete, but the *GmNMHC5* overexpression plants had already completed pollen formation (Figure 2).

### 2.3. Targeted Mutagenesis of Gmnmhc5 Induced by CRISPR-Cas9

*GmNMHC5* was edited by the editing tool CRISPR-Cas9. One target site (named *GmNMHC5-SP1*) in the second exon of *GmNMHC5* was chosen (Figure 3a), and the corresponding sgRNA-Cas9 vector was transformed into the soybean cultivar Jack via *Agrobacterium tumefaciens* mediated transformation. In this experiment, we obtained 12 transformed plants (87 seeds) in the T0 generation, which were tested by sequencing including three edited strains (*Gmnmhc5-#5-1* to *Gmnmhc5-#5-3*). We separately planted individual edited plants. The T1 generation seeds were detected as for T0. After sequencing, we got successfully edited three lines plants (*Gmnmhc5-#5-1* to *Gmnmhc5-#5-3*) in these T1 homozygous lines. After sequencing comparison, we detected one mutation right at the target site *GmNMHC5*-*SP1* (112-bp insertion and 6-bp mutation) (Figure 3b,c). The type of frameshift mutations induced by CRISPR-Cas9 at the target site of *GmNMHC5* generated premature translation termination codons (PTCs) (Appendix A). Therefore, the *GmNMHC5* gene will not be translated and the function will be lost. We therefore chose these 3 lines for continuous breeding and subsequent phenotypic analyses. By planting *Gmnmhc5-#5-1* to *Gmnmhc5-#5-3* separately, we obtained a T2 generation (including 33 plants) with stable inheritance of editing type and conducted phenotypic analyses. In addition, we also tested the marker gene bar with a test strip, and the test strips were all positive (Appendix A), indicating that CAS9 cassette still exists in the mutant strain. Thus, the stable inheritance of induced mutation (*Gmnmhc5-#5-1* to *Gmnmhc5-#5-3*) was obtained.

### 2.4. Gmnmhc5 Showed Obviously Late Flowering in Soybean

In the T1 generation of *Gmnmhc5*, we compared the flowering time between the mutants and the WT and found that homozygous mutagenesis of *Gmnmhc5* at the target site delayed flowering time. The T2 progeny of homozygous *Gmnmhc5-#5-1* to *Gmnmhc5-#5-3* lines were grown under both LD and SD photoperiodic conditions. Under LD conditions, the T2 *Gmnmhc5* mutants did not have floral buds when WT plants had already flowered. When *Gmnmhc5* flowered, the WT plants began to produce pods (Figure 4a). Under SD conditions, the T2 *Gmnmhc5* mutants did not have floral buds when WT plants were flowering, and when *Gmnmhc5* began to flower, the WT plants had obvious pods (Figure 4b). The comparison of flowering time between T2 homozygous *Gmnmhc5* mutants and WT plants under both LD and SD conditions is shown in Figure 4d. In addition, the *Gmnmhc5* mutants also showed late maturation (Figure 4c,e). The fact that loss of function of *GmNMHC5* delayed flowering and maturation phenotype under both LD and SD conditions strongly suggests that *GmNMHC*5 is a positive factor of flowering.

### 2.5. Overexpression of GmNMHC5 Inhibited the Expression of FT1a and FT4

To further investigate the molecular mechanism of *GmNMHC5* on flowering, we used transcriptome sequencing (RNA-Seq) to explore differential gene expression in response to overexpression of *GmNMHC5.* Compared with the WT, there were 4414 and 1522 differentially expressed genes (DEGs) in the two independent transformation events (line 32 and line 25) (Appendix A). Among them, there were 1081 genes with common differences between two independent transformation events (Appendix A). To ensure the reliability of the data, we only used these 1081 DEGs in the following analysis. Comparative analysis using the Phytozome (https://phytozome.jgi.doe.gov/pz/portal.html) and Uniprot (https://www.uniprot.org/) databases indicated that at least 22 overlapping DEGs (including *GmNMHC5*) showed homology with known flowering time-associated genes from *Arabidopsis*. The heat map of the differentially expressed genes in the transcriptome data and the results of GO enrichment analysis are uploaded to the Appendix A. In the two parts of up-regulation and down-regulation, the top ten pathways with the most DIGs were selected, and the specific contents of these analyses were listed in the Appendix A.

Two flowering inhibition genes, *GmFT1a* and *GmFT4*, were severely suppressed by overexpression of *GmNMHC5*, while the key promotion genes, *GmFT2a* and *GmFT5a*, in the classical flowering pathway showed no clear up or down trend in the 35S:*GmNMHC5* transgenic lines, implying that the overexpression of *GmNMHC5* might cause early flowering in soybeans through inhibiting the flowering suppressor genes. To further confirm this conjecture, we performed qRT-PCR assays with leaf samples to verify the expression of the four genes (*GmFT1a, GmFT2a, GmFT4*, and *GmFT5a*) identified from the RNA-Seq analysis (Figure 5). The results suggested that the expressions of these genes were consistent with the RNA-Seq results. The primers used in this section are also shown in Appendix A.

### 2.6. GmNMHC5 Could Rescue the Extremely Late Flowering of ft-10 in Arabidopsis

In order to further explore the relationship between *GmNMHC5* and *FT* in the flowering regulatory network, *GmNMHC5* was overexpressed in *ft-10* of *Arabidopsis*. The late flowering phenotype of *ft-10* was rescued after heterologous expression of *GmNMHC5* (Figure 6a). The flowering time of transgenic plant 35S:*GmNMHC5-ft-10* was basically the same as that of wild *Arabidopsis thaliana* (Figure 6b), revealing the functional complementarity effect of *GmNMHC5* on *FT*. This result provided a basis for further study on the positioning of *GmNMHC5* in the flowering network, especially in the regulatory relationship with *FT*.

## 3. Discussion

The gene *GmNMHC5* belongs to the MADS-box family. The MADS-box family has been confirmed as important transcription factors (TFs) involved in multiple stages of plant growth and development. Previous studies have revealed that they participate in development of angiosperm flower organs [27], in regulating the time of flowering initiation and tissue differentiation [28,29], and in the regulation of pollen [30] and fruit development [31,32]. Moreover, the MADS-box family also plays an important role in plant root development. For example, *AGL19* has been shown to be specifically expressed in root meristem and central column cells of mature roots in *Arabidopsis* [33]. Our research group also confirmed that *GmNMHC7* inhibits nodulation [9]. In addition to the genes that regulate flowering and root development alone, there are also some genes that are involved in both flowering and root development, such as *SHP1*, *SHP2*, *STK*, *AGL20* [34,35], and *AGL12* [36,37], highlighting bi-functional genes. Our previous research showed that *GmNMHC5* promotes soybean and lateral root development and nodule formation [13]. In this study, overexpressing this gene promoted flowering under both LD and SD, while using the CRISPR-Cas9 to edit *GmNMHC5* delayed flowering, showing that *GmNMHC5* is a soybean flowering promoting factor. These results indicate that *GmNMHC5* has dual functions in the regulation of flowering and in the nodulation process.

In *Arabidopsis*, *SUPPRESSOR OF OVEREXPRESSION OF CO1* (*SOC1*) (also a member of the MADS-box family), a very important TF that integrates the signaling from the gibberellin (GA)-dependent pathway, can induce flowering in non-flowering mutants [38]. Overexpression of *SOC1* promotes flowering under both LD and SD, while *soc1* mutants exhibit late flowering [5,39]. Since a similar regulatory function was observed in *GmNMHC5*, we can infer that *GmNMHC5* may have a similar regulatory pattern as *SOC1*. Since *GmNMHC5* and *SOC1* have the same pattern in the regulation of flowering, we believe that *GmNMHC5* has the same important function as *SOC1* in the regulation pathway of flowering. This conjecture adds a new element for further study of the flowering regulation network of soybean.

*FT* is the key point to integrate signals from various flowering pathways to regulate flowering. Therefore, the relationship between relevant genes and *FT* in the flowering regulation network has gradually become a research hotspot. The flowering promotion of *FT*, as a component of florigen, is conserved in many plants [40]. In recent years, it has been found that the *FT* gene differentiated in the process of evolution, resulting in its ability to inhibit flowering. For example, in sugar beet (*Beta vulgaris* L.), there are two *FT* homologous genes, *BvFT1* and *BvFT2*, which have completely opposite regulatory effects on flowering [41]. Similarly, Kong et al. found at least ten *FT* genes in the soybean, among which *GmFT2a* and *GmFT5a* were induced under SD and could interact with GmFDL19 to promote soybean flowering [20,21]. After targeted editing by CRISPR-Cas9, the *GmFT2a* mutants showed obviously late flowering [22]. Therefore, the potential flowering inhibition of *GmFT1a* and *GmFT4* proves that *FT* genes in the soybean also have functional differentiation, that is, they jointly control the direction of plant growth.

In this study, transcriptome analysis showed that the expression levels of *GmFT1a* and *GmFT4* in *GmNMHC5* overexpressed transgenic lines were significantly lower than those in the wild type, while there was no significant difference between *GmFT2a* and *GmFT5a*. These results were also confirmed by QT-PCR. Therefore, we hypothesized that *GmNMHC5* promoted flowering and inhibited the flowering-inhibiting genes. As there is currently no further evidence to prove a clear causal relationship between the two, the conjecture has certain limitations, suggesting a direction for future research.

However, experiments in which *GmNMHC5* was overexpressed in *ft-10* suggested that *GmNMHC5* could rescue the function of the deficient *FT*, so we speculated that *GmNMHC5* was located downstream of *FT* in the flowering pathway of *Arabidopsis*. This seems contradictory, but there can only be one explanation, that is, they are located in two different regulatory pathways and thus do not have a simple upstream and downstream relationship; parallel pathways likely exist. In *Arabidopsis*, Yamaguchi et al. have shown that *TWIN SISTER OF FT* (*TSF*) is located in a specific flowering regulation pathway other than that carried out by *FT*, despite it also being a homologous gene of *FT* [42]. Therefore, we speculate that *GmFT1a* and *GmFT4* (both flowering suppressor genes) and the promotion genes (*GmFT5a* and *GmFT2a*) are similar to the regulatory pattern of TSF and FT in *Arabidopsis* suggested by Yamaguchi et al. Namely, the two gene groups also show a parallel regulatory relationship in soybean flowering. Based on the *GmNMHC5* regulation, this study also elucidates the regulation mode of this group of homologous *FT* genes, providing a new idea for further research on the localization of the ten members of *FT* homologs in the soybean in the flowering regulation network.

The soybean is sensitive to photoperiod, and switches from vegetative growth to reproductive growth only after the sunshine length is shortened to a certain limit. A cultivar can thus only be grown in a limited area. The discovery of the function of *GmNMHC5* in regulating flowering in this study is expected to be used in breeding practice to coordinate with other genes that control flowering and adjust the photoperiod sensitivity of soybeans so as to expand or adjust the applicable area of soybean varieties. Further research on this gene is likely to have important guiding significance in future breeding work.

## 4. Material and Methods

### 4.1. Plant Materials and Growth Conditions

In this study, the soybean (*Glycine max* (L.) Merr.) cultivar Jack was selected as the wild type. Non-transgenic strains of Jack were used as the control group, and denoted WT. The experimental groups contained 35S:*GmNMHC5* mutant plants (overexpression of the *GmNMHC5* gene) and *Gmnmhc5* plants (targeted mutagenesis of *GmNMHC5* induced by CRISPR-Cas9), constructed from the wild-type Jack. DNA extracted from leaf tissue was used to examine the CRISPR-Cas9-induced mutations at the target site using PCR and DNA sequencing analysis. The T0 transgenic lines containing the T-DNA of the sgRNA-Cas9 vectors were identified, and then, all collected seeds from these self-pollinated T0 lines were planted under natural conditions (summer) in Beijing, China. Site-directed mutagenesis of *Gmnmhc5* was also observed at the target site in the T1 generation. Seeds of WT, 35S:*GmNMHC5* (from T1 generation), and *Gmnmhc5* (from T1 generation) homozygous mutants were separately grown in a controlled culture room under long-day (LD, 16 h light/8 h dark) and short-day (SD, 12 h light/12 h dark) photoperiodic conditions at 27 °C with 50% relative humidity. After screening, we successfully obtained stable genetic overexpression lines 35S:*GmNMHC5#5*, 35S:*GmNMHC5#25*, 35S:*GmNMHC5#32*, and *Gmnmhc5#5* (edited by CRISPR-Cas9).

The seeds of *Arabidopsis* were disinfected using sodium hypochlorite for 15 min, and rinsed four to five times with ddH_2_O. They were seeded onto 1/2 MS medium and incubated at 4 °C for 2–3 d, then transferred to 22 °C in a light incubator and cultured (16 h light per day). When the *Arabidopsis* grew to 4–6 leaves, they were planted in the vermiculite/nursery substrates soil (1:1) mixture, covered with cling film, and grown for 3 to 5 d (22 °C, 16 h light). To screen for transgenic *Arabidopsis* 35S:*GmNMHC5-ft10*, selected seeds (T1 generation) were planted on 1/2 MS medium containing screening agent hygromycin (hpt, Roche^®^, Basel, Switzerland) [43]. The positive transgenic *Arabidopsis* plants were further screened by sequencing. The primers used for sequencing are listed in Appendix A. The T2 generation seeds were first cultured on the screening medium and then transferred to 1/2 MS medium. Seeds after the T3 generation were directly planted on 1/2 MS medium for the subsequent experiments.

### 4.2. Time Measurements and Statistical Analyses

Soybean flowering time is calculated based on the time from days after emergence (DAE) to the R1 stage when the first flower is present on any node. In order to eliminate any defects in the formation and development of floral organs, we also uploaded a picture of wild-type flowering as a control in Appendix A. The maturation time was confirmed from the day when 95% of the pods had reached maturity (R8 Stage) [44]. For phenotypic statistics of flowering time, at least 13 plants were analyzed for each genotype of soybean. Microsoft Excel was used for statistical analyses. A one-way analysis of variance (ANOVA) was used to compare the significance of differences between controls and treatments at the 0.01 probability level. Origin 2017 (https://www.originlab.com/) was used for drawing the figures.

### 4.3. Paraffin Sectioning

The top buds of soybean seedlings in the same growth period were selected, dissected with an anatomical needle and fixed in formaldehyde-acetic acid-ethanol (FAA) for 24 h. The treatments were as follows: 50%, 70%, and 85% alcohol solutions were used to dehydrate each tissue for 30 min, samples were then placed in 95% alcohol containing 0.5% eosin for 3 h, and anhydrous alcohol for 1 h (2 times); the alcohol solutions of 25%, 50%, and 75% were used for xylene transparentizing for 30 min, and 100% xylene for 30 min (3 times). After treatment in 50% xylene + 50% paraffin, samples were embedded within pure wax in a 60 °C oven. The samples were sliced using a Leica RM 2235 slicer, displayed and baked in an oven. Then, slices were dewaxed with xylene, alcohol, and distilled water, mordanted with 4% iron alum for 30 min, dyed with 0.5% hematoxylin, and finally treated with 2% alum for color separation. After washing and returning to blue, slices were dehydrated with alcohol solution. After being redyed in 95% alcohol containing 1% eosin, and after being sealed, the slices were observed under an Olympus BX 51 optical microscope. Photoshop was used to prepare the pictures [45].

### 4.4. sgRNA Design and Construction of the CRISPR-Cas9 Expression Vector

A plasmid vector carrying both sgRNA and Cas9 cassettes was constructed. The Cas9 sequence was codon-optimized and assembled downstream of the CaMV2 35S promoter, together with the specific sgRNA driven by the Arabidopsis U6 promoter. The bar gene was used as a screening marker. Information on the gene *GmNMHC5* was from the Phytozome (https://phytozome.jgi.doe.gov/pz/portal.html). sgRNA was designed by using the CRISPR-P (http://cbi.hzau.edu.cn/crispr/) [46] with 5′-NGG (PAM, protospacer adjacent motif) in the strand. In this study, an sgRNA with *GmNMHC5* as the target was finally selected, and we named it *Gmnmhc5-SP1.*

### 4.5. Transformation of CRISPR-Cas9 in Soybeans and Screening for Mutations by Sequencing Analysis

In the transformation experiment, the CRISPR-Cas9 expression vector was transferred to *Agrobacterium tumefaciens* EHA105 by electroporation, and the soybean variety Jack was selected for tissue culture and transformation [47]. In order to confirm the mutations, plants were screened by dabbing leaves with 160 mg/L glufosinate solution, and were genotyped for the presence of the transgene using PCR. The leaves of each plant were collected, and genomic DNA was extracted from the leaves. Subsequently, the regions around the target site were amplified by PCR using Phanta^®^ Super Fidelity DNA Polymerase (Vazyme Biotechnology, Nanjing, China) with forward (5-CCAGCCATCCTCTTGCGTTA-3) and reverse primers (5-ATGCTTGGGAAGTCGGAAGG-3) of *GmNMHC5.* Sequencing was performed to determine whether the edit was successful. The type of gene editing can be determined by sequencing a peak map. CRISPR-Cas9-induced base insertions or deletions (not multiples of 3) can eventually result in transcoding mutations that cause a loss of gene function. By analyzing the peak diagram of sequencing results, overlapping peaks from the target site to the end of the sequence were considered a heterozygous mutation with successful editing. However, the wild type and unedited lines would not have any overlapping peaks. Then, homozygous mutants were determined based on sequence comparison with the wild type. T1, T2, and T3 plants were also tested by this method.

### 4.6. Transcriptome Analysis and Gene Functional Annotation

For the transcriptome analysis, two biological replicates of transgenic mutants were analyzed. We used wild-type Jack as the control group, and 35S:*GmNMHC5#25* and 35S:*GmNMHC5*-#32 as the experimental groups. The materials were grown in a controlled culture room at 28 °C under LD conditions, and leaf samples were taken at 31 DAE. Each sample consisted of material collected from three individual plants. All collected tissues were frozen immediately in liquid nitrogen and stored at −80 °C. mRNA extracts from the samples were sequenced with the HiSeq 4000 platform (Illumina, San Diego, CA, USA) following the manufacturer protocols. Raw data were initially processed using in-house Perl scripts. In this step, clean reads were obtained by removing those containing adapters. The clean reads were aligned to the soybean reference genome using TOPHAT v.2.0.9 (tophat.cbcb.umd.edu/). HTSEQ v.0.5.4p3 (https://htseq.readthedocs.io/en) was used to count the read numbers mapped to each gene. Then, the fragments per kilobase of transcript per million mapped reads of each gene were calculated based on the length of the gene and fragment count mapped to this gene. Differential expression analysis was conducted using the DES_EQ_ R package (v.1.10.1) (http://bioconductor.org/packages/2.11/bioc/html/DESeq.html), and the *p*-value results were adjusted using the Benjamini–Hochberg method to control for the false discovery rate. Genes with adjusted *p* < 0.05 were considered as differentially expressed genes (DEGs).

### 4.7. Gene Expression Validation by qRT-PCR

AnABI7900 thermocycler (Applied Biosystems, Foster City, CA, USA) and Takara SYBR premix Ex Taq (Takara, Kusatsu, Japan) were used for quantitative RT-PCR (qRT-PCR). A total of three biological replicates and three technical replicates were used. Microsoft Excel was used to analyze the qRT-PCR data. Appendix A shows the primers used in the experiment. The samples of gene expression detection were collected at 12:00 AM, selecting the expanded trifoliate leaves of 31-DAE soybeans under long day conditions. The internal reference gene is actin. A total of 34 cycles were used for RT-PCR analysis.

### 4.8. Accession Numbers

The clean data of the RNA-seq were deposited in the SRA database of NCBI under the accession number PRJNA635458.

## Figures and Tables

**Figure 1 plants-09-00792-f001:**
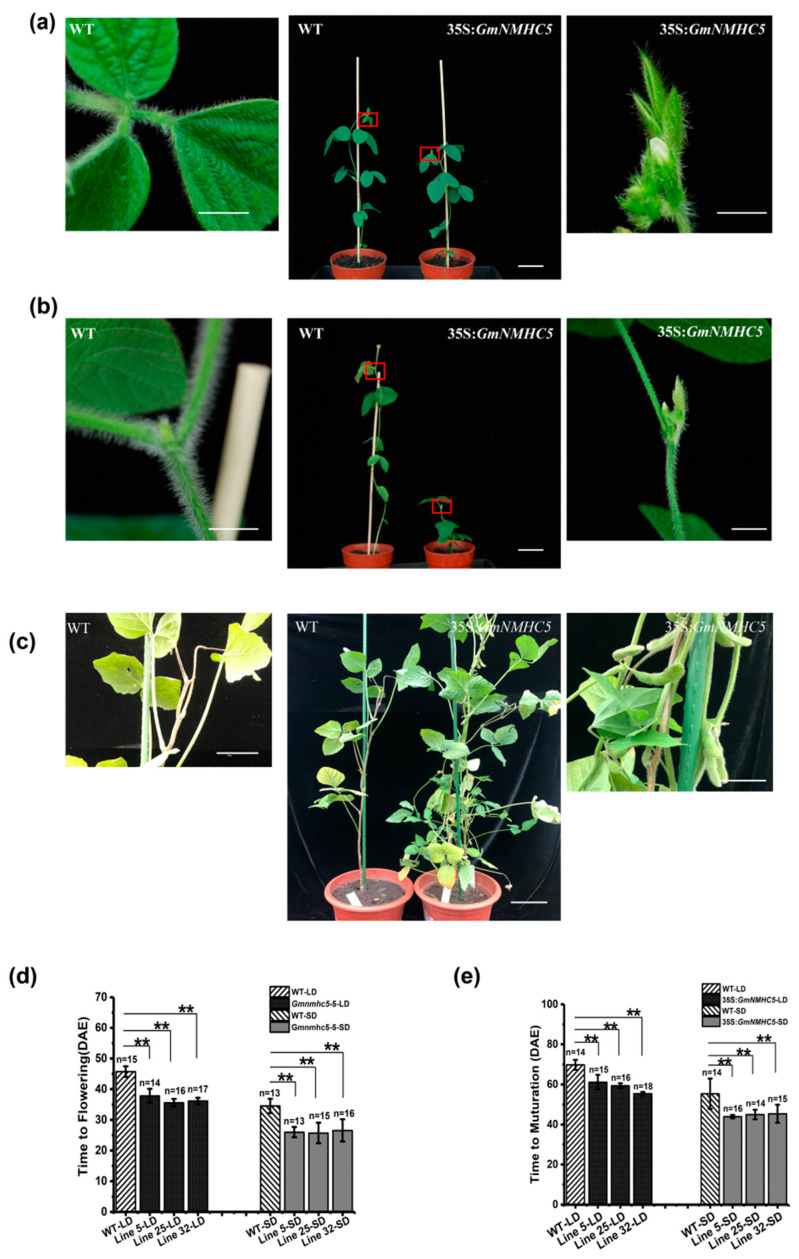
Phenotypes of the *GmNMHC5* transgenic soybean plants. (**a**) An overview of WT soybean plants, 35S:*GmNMHC5* at 37 days after emergence (DAE) under long-day (LD) conditions, and a close-up view of the flower areas framed by the boxes. (**b**) An overview of WT soybean plants, 35S:*GmNMHC5* at 25 DAE under short-day (SD) conditions, and a close-up view of the flower areas framed by the boxes. (**c**) An overview of WT soybean plants, 35S:*GmNMHC5* at 48 DAE under LD conditions, and a close-up view of the areas framed by the boxes. (**d**) Flowering times of 35S:*GmNMHC5* mutants and WT plants. The exact numbers of individual plants are shown. Under LD conditions: WT (*n* = 15), line 5 (*n* = 14), line 25 (*n* = 16), line 32 (*n* = 17); under SD conditions: WT (*n* = 13), line 5 (*n* = 13), line 25 (*n* = 5), line 32 (*n* = 16). A one-way analysis of variance (ANOVA) was used to compare the significance: **, *p* < 0.01. (**e**) Maturation times of 35S:*GmNMHC5* and WT plants. The exact numbers of individual plants are shown. Under LD conditions: WT (*n* = 15), line 5 (*n* = 14), line 25 (*n* = 16), line 32 (*n* = 18); under SD conditions: WT (*n* = 14), line 5 (*n* = 16), line 25 (*n* = 14), line 32 (*n* = 15). A one-way analysis of variance (ANOVA) was used to compare the significance: **, *p* < 0.01. Scale bar: 10 cm.

**Figure 2 plants-09-00792-f002:**
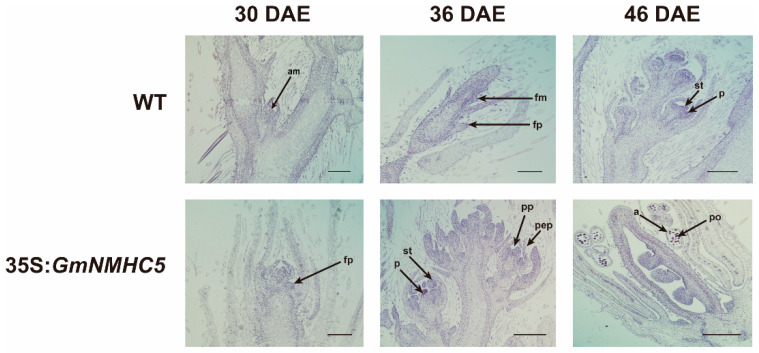
Flower structure development of wild-type and 35S:*GmNMHC5* strains displayed in paraffin sections. All plants were treated under LD conditions. Apical raceme primordium appeared at 30 DAE in the transgenic lines, while the wild-type plants were still in the vegetative growth stage. A terminal floral bud of 35S:*GmNMHC5* mutant was observed at 36 DAE while the flower bud structure of wild type had just appeared. Similar structures were found at 46 DAE in WT. (Abbreviations, a: Anther, am: Apical meristem, c: Carpel, fp: Floral primordium, fm: Floral meristem, po: Pollen, pp: Pistil primordium, st: Stamen, pep: Petal primordium, and p: Pistil). Scale bar: 200 μm.

**Figure 3 plants-09-00792-f003:**
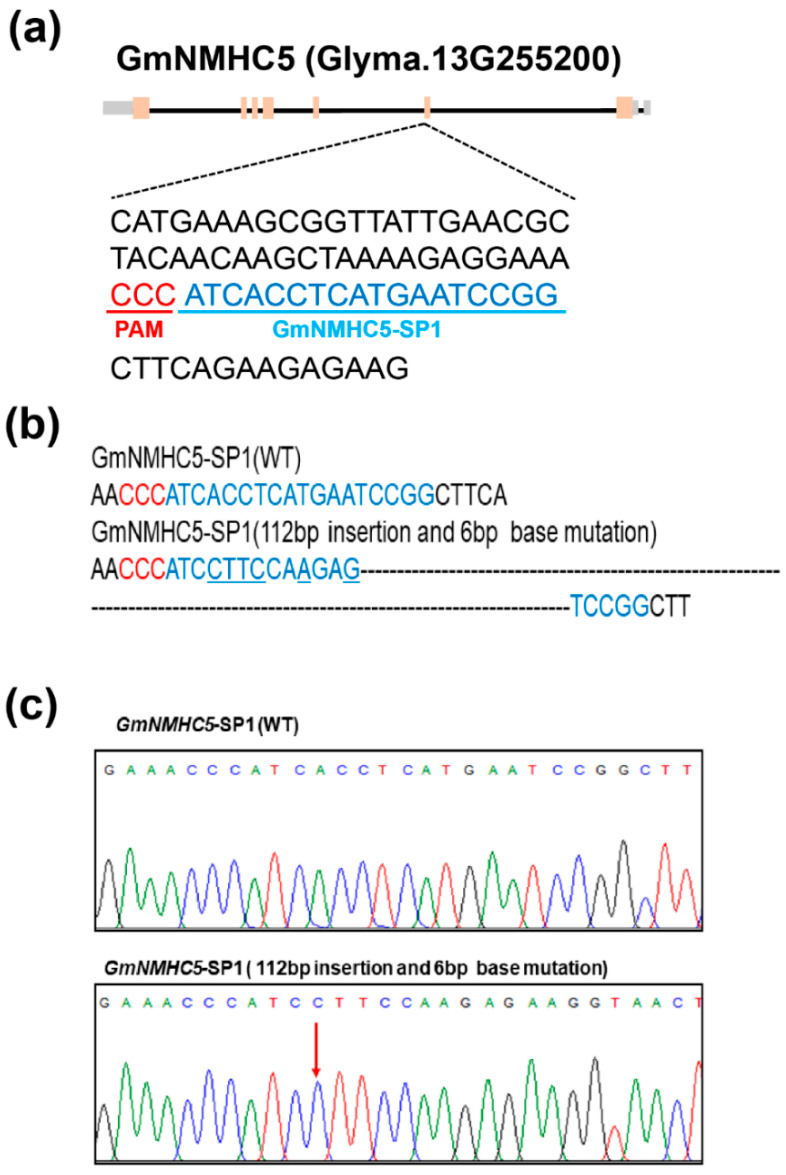
(**a**) Gene structure of *Gmnmhc5* with target site of CRISPR-Cas9 designed in the second exon. Pink stripe: Exon. Black line: Intron. Gray stripe: Untranslated regions (UTR). The underlined nucleotides indicate the target sites (named *GmNMHC5*-*SP1*). Nucleotides in red represent protospacer adjacent motif (PAM) sequences. (**b**) Sequences of wild type and representative mutation type induced at target site *GmNMHC5*-*SP1* are presented (mutations and insertions). (**c**) Sequence peaks of wild type and representative mutation type at target site *GmNMHC5*-*SP1*. The red arrow indicates the beginning location of mutations.

**Figure 4 plants-09-00792-f004:**
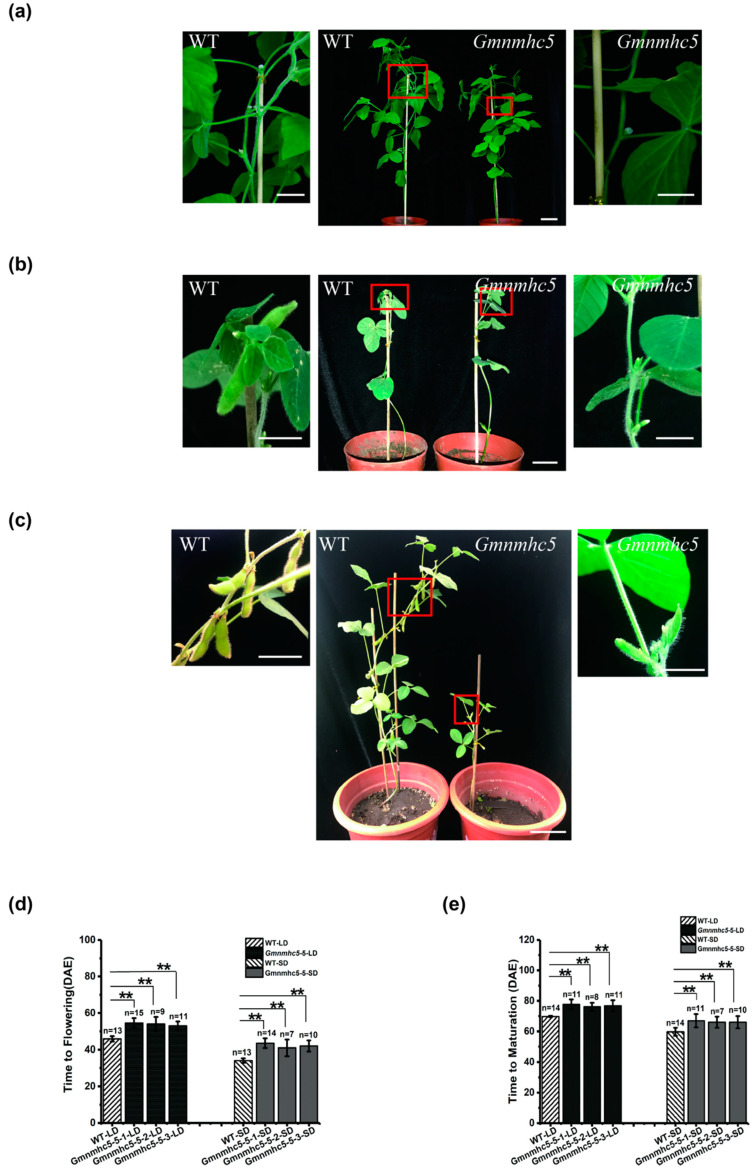
Phenotypes of the CRISPR-Cas9-induced *Gmnmhc5* soybean plants. (**a**) An overview of WT soybean plants and homozygous T2 *Gmnmhc5* at 54 DAE under LD conditions, and a close-up view of the flower areas framed by the boxes. (**b**) An overview of WT soybean plants and homozygous T2 *Gmnmhc5* at 44 DAE under SD conditions, and a close-up view of the flower areas framed by the boxes. (**c**) An overview of WT soybean plants and homozygous T2 *Gmnmhc5* at 62 DAE under LD conditions, and a close-up view of the areas framed by the boxes. (**d**) Flowering times of WT and *Gmnmhc5* plants. The exact numbers of individual plants are shown. Under LD conditions: WT (*n* = 13), *Gmnmhc5-5-1* (*n* = 15), *Gmnmhc5-5-2* (*n* = 9), *Gmnmhc5-5-3* (*n* = 11); under SD conditions: WT (*n* = 13), *Gmnmhc5-5-1* (*n* = 14), *Gmnmhc5-5-2* (*n* = 7), *Gmnmhc5-5-3* (*n* = 10). A one-way analysis of variance (ANOVA) was used to compare the significance: **, *p* < 0.01. (**e**) Maturation times of WT and *Gmnmhc5* plants. The exact numbers of individual plants are shown. Under LD conditions: WT (*n* = 14), *Gmnmhc5-5-1* (*n* = 11), *Gmnmhc5-5-2* (*n* = 8), *Gmnmhc5-5-3* (*n* = 11); under SD conditions: WT (*n* = 14), *Gmnmhc5-5-1* (*n* = 11), *Gmnmhc5-5-2* (*n* = 7), *Gmnmhc5-5-3* (*n* = 10). A one-way analysis of variance (ANOVA) was used to compare the significance: **, *p* < 0.01. Scale bar: 5 cm.

**Figure 5 plants-09-00792-f005:**
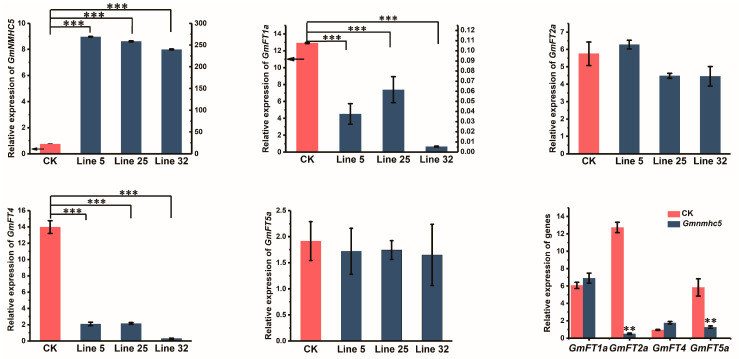
Expression of the selected flowering-related genes by qRT-PCR. Expression levels of *GmNMHC5*, *GmFT1a*, *GmFT4*, *GmFT2a*, and *GmFT5a* in leaves at 31 DAE under LD conditions were measured. The relative expression levels are normalized to *GmActin*. CK stands for wild-type plants, and the data of the CK histogram in panels 1 and 2 correspond to the value on the left coordinate axis (arrow pointing). The data are means ± SE of three biological replicates. Statistical significance was determined using a one-way analysis of variance (ANOVA): ** *p* < 0.01, *** *p* < 0.001.

**Figure 6 plants-09-00792-f006:**
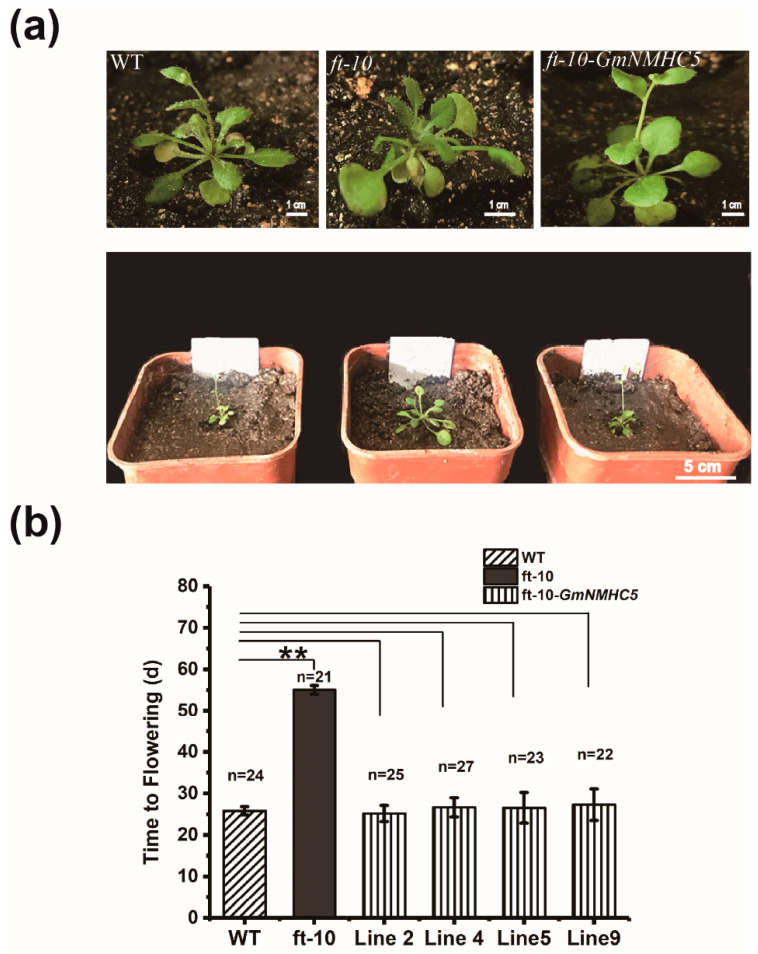
Phenotypes of the 35S:*GmNMHC5-ft-10* transgenic *Arabidopsis* plants. (**a**) An overview of wild-type *Arabidopsis* (Col-0) plants and 35S:*GmNMHC5-ft-10* at 28 d under LD conditions (upper panel), and a close-up view of the areas framed by the boxes (lower panel). (**b**) Flowering times of WT, *ft-10*, and 35S:*GmNMHC5-ft-10-2*, 35S:*GmNMHC5-ft-10-4*, 35S:*GmNMHC5-ft-10-5*, and *35S:GmNMHC5-ft-10-9* plants under LD conditions. The exact numbers of individual plants are ≥ 21. A one-way analysis of variance (ANOVA) was used to compare the significance: **, *p* < 0.01.

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
