# Peer review of "GmNMHC5, A Neoteric Positive Transcription Factor of Flowering and Maturity in Soybean"

_plants, 2020, doi:10.3390/plants9060792_

Round 1

Reviewer 1 Report

In the present work submitted to Plants, Wu and colleagues describes a new role of the MADs-box transcription factor GmNMHC5 in Glycine max. Authors have previously shown that GmNMHC5 have a function in root development and nodulation. In this new work, through generation and phenotypic and molecular characterization of loss of function CRISPR-Cas edited alleles and gain of function overexpression lines in soybean, and heterologous expression in the Arabidopsis thaliana ft-10 mutant, they show that GmNMHC5 has also a role in promoting transition to flowering and maturity in Glycine max.

The genetics and molecular experimental approach chosen to reveal this new role of the gene in the regulation of flowering and maturation is more than adequate. Experiments are, in general, well performed and documented. However I have many concerns that I have included directly in the text (comments, small revisions, corrections and requests for clarifications), and which need to be address by authors in order to improve the quality and understanding of the work. Please, check them all and correct when necessary. Among the clarifications and requests that I make to the authors, I must highlight here my two major concerns regarding procedures and results:

  1. Authors should address the inclusion on results that are not shown in the paper, mostly related with the RNA-Seq experiment. In this kind of analysis is important to know the identity of differentially-expressed genes (DIGs) and their fold-changes. I couldn´t find this information anywhere in the text or supplemental files. This information should be add as supplemental material. Moreover, in order to cover the whole range of biological processes related to the regulation exerted by GmNMHC5, a more thoroughly in silico analysis of the list of DIGs would be appreciated.
  2. The mutations caused by CRISPR-Cas editing should be correlated with changes in the protein. Authors should specify how the 7bp change and the 122 pb insertion in the nucleotide sequence will change the aminoacid sequence of the protein, and discuss how these changes will modify GmNMHC5 function.

Regarding the figures, some panels need lettering or specifications in the corresponding figure legends. Scale bars in every picture would be appreciated.

Reviewer 2 Report

In this manuscript, Wang and collaborators investigate the function of  GmNMHC5,  a soybean MADS box gene.

To this aim the generate several are transgenic lines. The authors:

  • overexpressed GmNMHC5,  using the 35S promoter
  • destroyed GmNMHC5, by using the CRISPR-CAS9 methodology

The transgenic plant phenotypes are coherent that is  that is 35S:GmNMHC5 plants anticipate flowering, whilst the individual edited show a late flowering phenotype.

The manuscript is of interest for the readership of Plants, nevertheless some revisions are needed before to accept for publications

  • About figure 1, I think it might be important to add a picture of a wild type flower at anthesis close to a flower produced by 35S:GmNMHC5 plants to exclude any defects of floral organ formation and development.

About 35S:GmNMHC5 plants, the authors describe a precocious senescence, I think it might be easier to follow such phenotype adding some pictures of old plants (indicating the age in days after germination). Which parameters did they analyse? What about pod maturation, is pod maturation anticipate too?

  • Similar comments must be considered for the pictures describing the CRISPR-CAS9 pants edited.

            About these plants is not clear whether the CAS9 is still present or not.

The statistical tests used to evaluate the significance of the mutant phenotypes is unclear. This needs to be well described in the figure legends including the number of samples analysed, the test used and the significance threshold used. I would also suggest performing a multiple testing correction since they are examining the effects of many individuals simultaneously.

  • The RNAseq data should be stored in a public database to be shared with the scientific community, the information about the database should be added. It could be useful to include the heat maps and GO analysis of the differentially expressed genes. What about cell cycle genes? Their miss regulation might be involved in the precocious senescence

Figure 2 should be improved, I suggest a toluidine blue staining  in place of eosin

Round 2

Reviewer 1 Report

Authors have properly address most of my demands, so the paper is now improved. However, I have still the next concerns:

  1. It is not entirely clear to me, by Text S1 (better Figure S1), which are the predicted changes in the coding sequence of the mutant allele. Is the data regarding the mutant CDS in figure S1, a prediction or comes from cDNA sequencing of the mutant transcript? I guess is the first. If so, I cannot fully understand how a 122 pb insertion in the second exon can remove the original start codon of the CDS, so I doubt the new predicted aminoacid sequence is the one shown. Please, check it or clarify. In my opinion, this additional figure in its present form is confusing. Maybe, would help if authors used three different colours for the the 122 pb insertion, the 7 pb changes and the original sequences in the whole mutant CDS.
  2. I also have some doubts about the size of the scale bar in figure 6. If it is in fact 4cm, that would mean that the rossetes of the arabidopsis plants would be around 9 cm diameter, and I don´t think so.
  3. Laslty, please do not use the term “text” to refer to the new supplemental figures and tables.

Reviewer 2 Report

The present manuscript improved but two points need some attention

  • The CAS9 cassette. I asked to verify whether the cassette is still present, this information must be added.
  • The statistical assays. The t-Student is not the correct one, One-Way ANOVA is a parametric test and it is more indicated for such analysis. The One-Way ANOVA ("analysis of variance") compares the means of two or more independent groups in order to determine whether there is statistical evidence that the associated population means are significantly different.
